# QUANTILE-LSTM: A ROBUST LSTM FOR ANOMALY DETECTION IN TIME SERIES DATA

## ABSTRACT

Anomalies refer to the departure of systems and devices from their normal behaviour in standard operating conditions. An anomaly in an industrial device can indicate an upcoming failure, often in the temporal direction. In this paper, we contribute to: 1) multiple novel LSTM architectures, q-LSTM by immersing quantile techniques for anomaly detection. 2) a new learnable, parameterized activation function Parameterized Elliot Function (PEF) inside LSTM, which saturates late compared to its nonparameterized siblings including the *sigmoid* and *tanh*, to model temporal long-range dependency. The proposed algorithms are compared with other well-known anomaly detection algorithms and are evaluated in terms of performance metrics, such as Recall, Precision and F1-score. Extensive experiments on multiple industrial timeseries datasets (Yahoo, AWS, GE, and machine sensors, Numenta and VLDB Benchmark data) and non-time series data show evidence of effectiveness and superior performance of LSTM-based quantile techniques in identifying anomalies.

## 1 INTRODUCTION

Anomalies indicate a departure of a system from its normal behaviour. In Industrial systems, they often lead to failures. By definition, anomalies are rare events. As a result, from a Machine Learning standpoint, collecting and classifying anomalies pose significant challenges. For example, when anomaly detection is posed as a classification problem, it leads to extreme class imbalance (data paucity problem). Morales-Forero & Bassetto (2019) have applied a semi-supervised neural network, a combination of an autoencoder and LSTM, to detect anomalies in the industrial dataset to mitigate the data paucity problem. Sperl et al. (2020) also tried to address the data imbalance issue of anomaly detection and applied a semi-supervised method to inspect large amounts of data for anomalies. However, these approaches do not address the problem completely since they still require some labeled data. Our proposed approach is to train models on a normal dataset and device some post-processing techniques to detect anomalies. It implies that the model tries to capture the normal behavior of the industrial device. Hence, no expensive dataset labeling is required. Similar approaches were tried in the past. Autoencoder-based family of models uses some form of thresholds to detect anomalies. For example, Sakurada & Yairi (2014); Jinwon & Ch (2015) mostly relied on reconstruction errors. The reconstruction error can be considered as an anomaly score. If the reconstruction error of a datapoint is higher than a threshold, then the datapoint is declared as an anomaly. However, the threshold value can be specific to the domain and the model, and deciding the threshold on the reconstruction error can be cumbersome.

MOTIVATION AND CONTRIBUTION

Unlike the above, our proposed quantile-based thresholds applied in the quantile-LSTM are generic and not specific to the domain or dataset. We have introduced multiple versions of the LSTM-based anomaly detector in this paper, namely (i) quantile-LSTM (ii) iqr-LSTM and (iii) Median-LSTM. All the LSTM versions are based on estimating the quantiles instead of the mean behaviour of an industrial device. For example, the median is $50\%$ quantile. Our contributions are three-fold:
**(1)** Introduction of Quantiles in design of quantile-based LSTM techniques and their application in anomaly identification.
**(2)** Proposal of the Parameterized Elliot as a 'flexible-form, adaptive, learnable' activation function

in LSTM, where the parameter is learnt from the dataset. We have shown empirically that the modified LSTM architecture with PEF performed better than the Elliot Function (EF) and showed that such behavior might be attributed to the slower saturation rate of PEF. PEF contributes to improved performance in anomaly detection in comparison to its non-parameterized siblings.

**(3)** Evidence of superior performance of the proposed Long Short Term Memory networks (LSTM) methods over state-of-the-art (SoTA) deep learning and non-deep learning algorithms across multiple Industrial and Non-industrial data sets including Numenta Anomaly Benchmark and the VLDB anomaly benchmark (Appendix, Table 7, 8, 9 and 10).

There are three key pieces to modelling anomalies: type of time-series we need to work with; model the temporal dependency and post-process the forecasts to flag that forecast as an anomaly. Given the nature of anomalies, it is obvious they should model the departure normality or the tail behaviour. Quantities are the natural statistical quantities to consider in this respect. The temporal modeling of time-series models is some sort of dynamical systems, including the classical statistical models like ARMA and its variants. LSTMs are the most popular versions of the non-parametric non-linear dynamical models. One could technically swap LSTMs with any other sequence architectures suitable for the problem. The added advantage LSTMs brings is the multiplicative gates which help prevent vanishing gradients. This is coupled with the introduction of Parameterized Elliot as activation function (PEF) which shifts the saturation. A classifier to flag anomalies is also a comparator, either learnt via supervised task or is based on reasonable heuristics. For the former, we need labels which we assume do not have in large numbers in reality, For the latter, there is no option but to default to some heuristics. But thankfully, with a non-parametric, non-linear dynamical system such as q-LSTM modelling the quantities, even fixed, deterministic comparators turn out to be adaptive comparators. Therefore, we can consider our contribution as setting this template and making certain sensible choices in each of the three important puzzles of this template. The rest of the paper is organized as follows. The proposal and discussion of various LSTM-based algorithms are presented in section 2. Section 3 describes the LSTM structure and introduces the PEF. This section also explains the intuition behind choosing a parameterized version of the AF and better variability due to it. Experimental results are presented in section 4. Section 5 discusses relevant literature in anomaly detection. We conclude the paper in section 6.

## 2 ANOMALY DETECTION WITH QUANTILE LSTMs

Quantiles are used as a robust alternative to classical conditional means in Econometrics and Statistics, as they can capture the uncertainty in a prediction and model tail behaviours (Koenker, 2005). The additional benefit lies in quantiles making very few distributional assumptions. It was also shown by Tambwekar et al. (2022) that quantiles aid in explainability as they can be used to obtain several univariate summary statistics that can be directly applied to existing explanation tools. This served as the motivation behind adopting the idea of quantiles from classification to anomaly detection, as quantiles capture tail behavior succinctly. It is well known that quantiles minimize check loss (Horowitz, 1992), which is a generalized version of Mean Absolute Error (MAE) arising from medians rather than means. It is also known that medians are often preferred to means in robust settings, particularly in skewed and heavy-tailed data. Thus, in time series data, where LSTM architecture has shown beneficial, LSTM architecture is coupled with the idea of quantiles to capture anomalies (outliers). It is to be noted that this method is applied to univariate time series data only, and the method is agnostic to data distribution (see Table 6 ). As the empirical results exhibit, the distributional variance does not impact the prediction quality.

Before we discuss quantile-based anomaly detection, we describe the data structure and processing setup, with some notations. Let us consider $x_i, i = 1, 2, .., n$ be the $n$ time-series training datapoints. We consider $T_k = \{x_i : i = k, \cdots, k + t\}$ be the set of $t$ datapoints, and let $T_k$ be split into $w$ disjoint windows with each window of integer size $m = \frac{t}{w}$ and $T_k = \{T_k^1, \cdots, T_k^w\}$. Here, $T_k^j = \{x_{k+m(j-1)}, ..., x_{k+m(j)-1}\}$. Let $Q_\tau(D)$ be the sample quantile of the datapoints in the set $D$. The training data consists of, for every $T_k$, $X_{k,\tau} \equiv \{Q_\tau(T_k^j)\}, j = 1, \cdots, w$ as predictors with $y_{k,\tau} \equiv Q_\tau(T_{k+1})$, sample quantile at a future time-step, as the label or response. Let $\hat{y}_{k,\tau}$ be the predicted value by an LSTM model.

## 2.1 Various quantile-LSTM Algorithms

A general recipe we are proposing to detect anomalies is to: (i) estimate quantile $Q_\tau(x_{k+t+1})$ with $\tau \in (0, 1)$ and (ii) define a statistic that measures the outlier-ness of the data, given the observation $x_{k+t+1}$. Instead of using global thresholds, thresholds are adaptive i.e. they change at every time-point depending on quantiles.

### 2.1.1 QUANTILE-LSTM

As the name suggests, in quantile-LSTM, we forecast two quantiles $q_{low}$ and $q_{high}$ to detect the anomalies present in a dataset. We assume the next quantile values of the time period after sliding the time period by one position are dependent on the quantile values of the current time period.

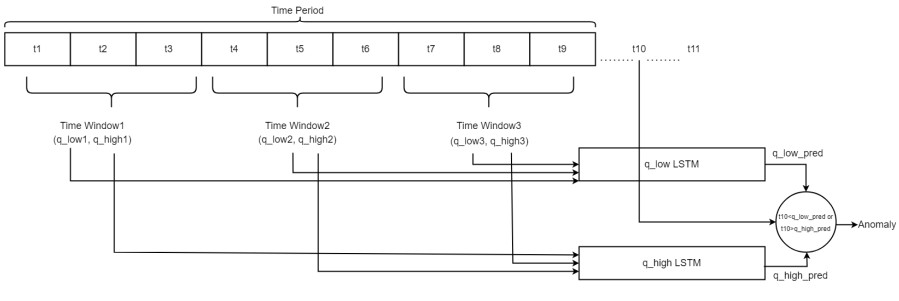

(a) Anomaly detection process using quantile-LSTM

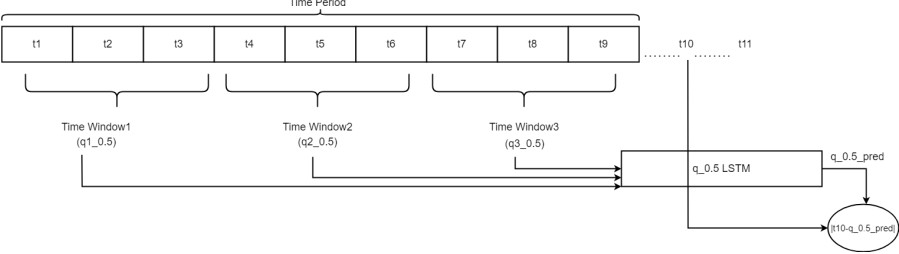

(b) Anomaly detection process using median-LSTM

Figure 1: Sigmoid function has been applied as an recurrent function, which is applied on the outcome of the forget gate ($f_t = \sigma(W_f * [h_{t-1}, x_t] + b_f)$) as well as input gate ($i_t = \sigma(W_i * [h_{t-1}, x_t] + b_i)$). PEF decides the information to store in cell $\hat{c}_t = PEF(W_c * [h_{t-1}, x_t] + b_c)$.

It is further expected that, nominal range of the data can be gleaned from $q_{low}$ and $q_{high}$. Using these $q_{low}$ and $q_{high}$ values of the current time windows, we can forecast $q_{low}$ and $q_{high}$ values of the next time period after sliding by one position. Here, it is required to build two LSTM models, one for $q_{low}$ (LSTM$_{qlow}$) and another for $q_{high}$ (LSTM$_{qhigh}$). Let's take the hypothetical dataset as a training set from Figure 1a. It has three time windows from time period $x_1 \cdots x_9$. Table 1 defines the three time windows of the time period $x_1 \cdots x_9$ and the corresponding $q_{low}, q_{high}$ values against the time window.

| TW | $q_{low}$ | $q_{high}$ |
|---|---|---|
| $x_1, x_2, x_3$ | $X_{1,low} \equiv Q_{low}(T_1^1)$ | $X_{1,high} \equiv Q_{high}(T_1^1)$ |
| $x_4, x_5, x_6$ | $X_{2,low} \equiv Q_{low}(T_1^2)$ | $X_{2,high} \equiv Q_{high}(T_1^2)$ |
| $x_7, x_8, x_9$ | $X_{3,low} \equiv Q_{low}(T_1^3)$ | $X_{3,high} \equiv Q_{high}(T_1^3)$ |

Table 1: The first time period and its corresponding time windows

The size of the inputs to the LSTM depends on the number of time windows $w$ and one output. Since three time windows have been considered for a time period in this example, both the LSTM models will have three inputs and one output. For example, the LSTM predicting the lower quantile,

would have $X_{1,low}, X_{2,low}, X_{3,low}$ as its puts and $y_{1,low}$ as its output, for one time-period. A total of $n - t + 1$ instances will be available for training the LSTM models assuming no missing values.

After building the LSTM models, for each time period it predicts the corresponding quantile value and slides one position to the next time period on the test dataset. Quantile-LSTM applies a following anomaly identification approach. If the observed value $x_{k+t+1}$ falls outside of the predicted $(q_{low}, q_{high})$, then the observation will be declared as an anomaly. For example, the observed value $x_{10}$ will be detected as an anomaly if $x_{10} < \hat{y}_{1,low}$ or $x_{10} > \hat{y}_{1,high}$. Figure 1a illustrates the anomaly identification technique of the quantile-LSTM on a hypothetical test dataset.

### 2.1.2 IQR-LSTM

IQR-LSTM is a special case of quantile-LSTM where $q_{low}$ is 0.25 and $q_{high}$ is the 0.75 quantile. In addition, another LSTM model predicts median $q_{0.5}$ as well. Effectively, at every time index $k$, three predictions are made $\hat{y}_{k,0.25}, \hat{y}_{k,0.5}, \hat{y}_{k,0.75}$. Based on this, we define the Inter Quartile Range (IQR) $\hat{y}_{k,0.75} - \hat{y}_{k,0.25}$. Using IQR, the following rule identifies an anomaly when $x_{t+k+1} > \hat{y}_{k,0.5} + \alpha(\hat{y}_{k,0.75} - \hat{y}_{k,0.25})$ or $x_{t+k+1} < \hat{y}_{k,0.5} - \alpha(\hat{y}_{k,0.75} - \hat{y}_{k,0.25})$.

### 2.1.3 MEDIAN-LSTM

Median-LSTM, unlike quantile-LSTM, does not identify the range of the normal datapoints; rather, based on a single LSTM, distance between the observed value and predicted median ($x_{t+k+1} - \hat{y}_{k,0.5}$) is computed, as depicted in Figure 1b, and running statistics are computed on this derived data stream. The training set preparation is similar to quantile-LSTM.

To detect the anomalies, Median-LSTM uses an implicit adaptive threshold. It is not reasonable to have a single threshold value for the entire time series dataset when dataset exhibits seasonality and trends. We introduce some notations to make description concrete. Adopting the same conventions introduced before, define $d_k \equiv x_{t+k+1} - Q_{0.5}(T_{k+1}), k = 1, 2, \ldots, n-t$ and partition the difference series into $s$ sets of size $t$ each, i.e., $D \equiv D_p, p = 1, \ldots, s$, where $D_p = \{d_i : i = (s-1)t + 1, \ldots, st\}$. After computing the differences on the entire dataset, for every window $D_p$, mean ($\mu_p$) and standard deviation ($\sigma_p$) are calculated for the individual time period $D_p$. As a result, $\mu_p$ and $\sigma_p$ will differ from one time period to another time period. Median-LSTM detects the anomalies using upper threshold and lower threshold parameters of a particular time period $D_p$ and they are computed as follows:

$$T_{p,lower} = \mu_p + w\sigma_p; T_{p,higher} = \mu_p - w\sigma_p$$

An anomaly can be flagged for $d_k \in T_p$ when either $d_k > T_{p,higher}$ or $d_k < T_{p,lower}$ Now, what should be the probable value for $w$? If we consider $w = 2$, it means that any datapoint beyond two standard deviations away from the mean on either side will be considered as an anomaly. It is based on the intuition that differences of the normal datapoints should be close to the mean value, whereas the anomalous differences will be far from the mean value. Hence 95.45% datapoints are within two standard deviations distance from the mean value. It is imperative to consider $w = 2$ since there is a higher probability of the anomalies falling into the 4.55% datapoints. We can consider $w = 3$ too where 99.7% datapoints are within three standard deviations. However, it may miss the border anomalies, which are relatively close to the normal datapoints and only can detect the prominent anomalies. Therefore we have used $w = 2$ across the experiments (See Appendix K for the characteristics of the proposed methods).

## 2.2 PROBABILITY BOUND

In this subsection, we analyze different datasets by computing the probability of occurrence of anomalies using the quantile approach. We have considered 0.1, 0.25, 0.75, 0.9, and 0.95 quantiles and computed the probability of anomalies beyond these values, as shown in Table 5 of Appendix A. The multivariate datasets are not considered since every feature may follow a different quantile threshold. Hence it is not possible to derive a single quantile threshold for all the features. It is evident from Table 5 of Appendix A that the probability of a datapoint being an anomaly is high if the datapoint's quantile value is either higher than 0.9 or lower than 0.1. However, if we increase the threshold to 0.95, the probability becomes 0 across the datasets. This emphasizes that a higher quantile threshold does not detect anomalies. It is required to identify the appropriate threshold value, and it is apparent from the table that most of the anomalies are nearby 0.9 and 0.1 quantile

values. Table 5 also demonstrates the different nature of the anomalies present in the datasets. For instance, the anomalies of Yahoo $Dataset_1$ to Yahoo $Dataset_6$ are present nearby the quantile value 0.9, whereas the anomalies in Yahoo $Dataset_7$ to Yahoo $Dataset_9$ are close to both quantile values 0.9 and 0.1. Therefore, it is possible to detect anomalies by two extreme quantile values. We can consider these extreme quantile values as higher and lower quantile thresholds and derive a lemma. We provide a proof in the Appendix B. The lemma entails the fact that anomalies are trapped outside the high and low quantile threshold values. The bound is independent of data distribution as quantiles assume nominal distributional characteristics.

# 3 LSTM WITH PARAMETERIZED ELLIOT ACTIVATION (PEF)

We introduce the novel parameterized Elliot activation function, which is the parameterized version of the Elliot activation function. The suitability of parameterized Elliot function, PEF, is conceived as an adaptive variant of usual activation wherein we modify the LSTM architecture by replacing the activation function of the LSTM gates with PEF. We save 'parameter tuning' efforts by learning the parameter values from backpropagation. Additionally, the cost of saturation of standard activation functions impedes training and prediction, which is an important barrier to overcome. We expect PEF to have a lower rate of saturation in LSTM compared to other activation functions such as tanh, sigmoid, etc. To the best of our knowledge, insights on 'learning' the parameters of an AF are not available in literature except for the standard smoothness or saturation properties AFs are supposed to possess. It is, therefore, worthwhile to investigate the possibilities of learning an AF within a framework or architecture that uses the inherent patterns and variances from data.

A single LSTM block is composed of four major components: an input gate, a forget gate, an output gate, and a cell state. We have applied the parameterized Elliot Function (PEF) as activation and introduced a parameter $\alpha$, which controls the shape of the Elliot function, represented by

$$f(x) = \frac{\alpha x}{1 + |x|} \tag{1}$$

with the first order derivative of PEF as: $f'(x) = \frac{\alpha}{(|x|+1)^2}$. The $\alpha$ in equation 1 is learnt during the back-propagation like other weight parameters of the LSTM model. There are multiple reasons to implement the PEF instead of other activation functions. Some of the reasons and salient features of the PEF are 1. The function is equal to 0, and the derivatives are also equal to $\alpha$ at the origin. 2. The function's derivative also saturates as the $|x|$ increases. However, the saturation rate is less than other activation functions, such as tangent. 3. One of the major benefits of the PEF is that it further decreases the rate of saturation in comparison to the non-parameterized Elliot function.

**Parameterized Elliot Function (PEF):** One of the major benefits of the Paramterized Elliot function is that it further decreases the rate of saturation in comparison to the non-parameterize Elliot function. We have introduced a parameter $\alpha$, which controls the shape of the Elliot function. After the introduction of the PEF, the hidden state equation is as follows: $h_t = O_t \alpha_c PEF(C_t)$. By chain rule, $\frac{\partial J}{\partial \alpha_c} = \frac{\partial J}{\partial \alpha_c} = \frac{\partial J}{\partial h_t} O_t * Elliot(C_t)$. After each iteration, the $\alpha_c$ is updated by gradient descent $\alpha_c^{(n+1)} = \alpha_c^n + \delta * \frac{\partial J}{\partial \alpha_c}$ (See Appendix C for back propagation of LSTM with PEF).

**Intuition behind Our Hypothesis:** Glorot & Bengio (2010) hypothesized that in neural networks, the logistic layer output softmax(b+Wh) might initially rely more on the biases b and hence push the activation value h towards 0, thus resulting in error gradients of smaller values. They referred to this as the saturation property of neural networks. This results in slower training and prevents the gradients from propagating backward until the layers close to the input learns. This saturation property is observed in the sigmoid. The sigmoid is non-symmetric around zero and obtains smaller error gradients when the sigmoid outputs a value close to 0. Similarly, tanh in all layers tends to saturate towards 1, which leads to layer saturation. All the layers attain a particular value, which is detrimental to the propagation of gradients. However, this issue of attaining saturation would be less pronounced in cases where two different activation functions are used. Since each activation function behaves differently in terms of gradients, i.e., sigmoid outputs are in the range [0,1], and the gradients are minimum at the maximum and minimum values of the function. Tanh, on the other hand, has minimum gradients at -1 and 1 and reaches its maximum at 0. Therefore, even if the layers begin to saturate to a common value, some of the layers would escape the saturation regime of their activations and would still be able to learn essential features. As an outcome, this might

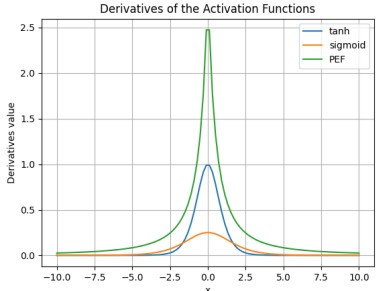

(a) Derivatives comparisons of various activation functions.

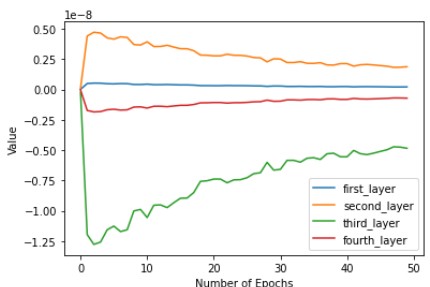

(b) LSTM values for 4 layers and 50 epochs using PEF as activation function using AWS2.

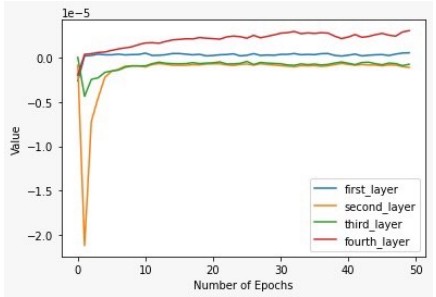

(c) LSTM values for 4 layers and 50 epochs using Sigmoid as activation function using AWS2.

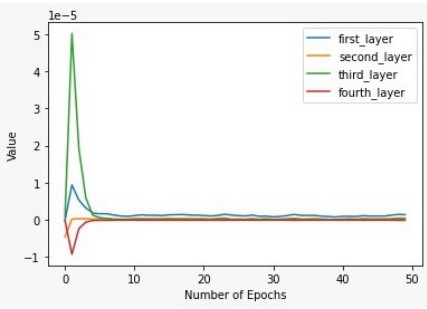

(d) LSTM values for 4 layers and 50 epochs using Tanh as activation function using AWS2.

| Dataset | $\alpha$ after training | $\alpha$ initial value |
|---------|------------------------|------------------------|
| AWS Dataset$_1$ | 1.612 | 0.1 |
| AWS Dataset$_2$ | 0.895 | 0.1 |
| AWS Dataset$_3$ | 1.554 | 0.1 |
| AWS DatasetSyn$_1$ | 1.537 | 0.1 |
| AWS DatasetSyn$_2$ | 0.680 | 0.1 |
| AWS DatasetSyn$_3$ | 1.516 | 0.1 |
| Yahoo Dataset$_1$ | 1.432 | 0.1 |
| Yahoo Dataset$_2$ | 1.470 | 0.1 |
| Yahoo Dataset$_3$ | 1.658 | 0.1 |
| Yahoo Dataset$_5$ | 1.686 | 0.1 |
| Yahoo Dataset$_6$ | 1.698 | 0.1 |
| Yahoo Dataset$_7$ | 1.725 | 0.1 |
| Yahoo Dataset$_8$ | 1.850 | 0.1 |
| Yahoo Dataset$_9$ | 1.640 | 0.1 |

Table 2: Different $\alpha$ values for each Dataset after the training.

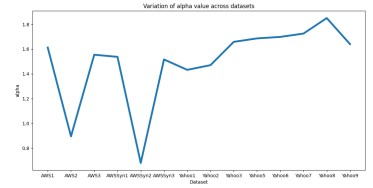

(e) The final $\alpha$ values learn on each dataset. We can see the final $\alpha$ value is different for different datasets.

Figure 2: The Fig shows the slow saturation rate as well as behavioral comparison of the different layers of LSTM model after the introduction of PEF with other activation functions. It also shows the final value of the learned parameter $\alpha$ on various datasets.

result in fewer instances of vanishing gradients. This assumption would mean that networks with two different activations would learn faster and converge faster to a minima, and the same premise is supported by a Convergence study (details in section V). As demonstrated by Glorot and Bengio, if the saturation ratio of layers is less pronounced, it leads to better results in terms of accuracy. A standard neural network with N layers is given by $h^l = \sigma(h^{l-1}W^l + b)$ and $s^l = h^{l-1}W^l + b$. Here $h^l$ is the output of the first hidden layer, $\sigma$ is a non-linear activation function, and b is the bias. We compute the gradients as $\frac{\partial Cost}{\partial s_k^l} = f'(s_k^l)W_{k,\cdot}^l \frac{\partial Cost}{\partial s^{l+1}}; \frac{\partial Cost}{\partial W_{m,n}^l} = z_l^i \frac{\partial Cost}{\partial s_k^l}$. Now, we find the variances of these values. As the network propagates, we must ensure that the variances are equal to keep the information flowing. Essentially, when $\forall(l, l'), Var[h^l] = Var[h^{l'}]$, it ensures that forward propagation does not saturate, and when $\forall(l, l'), Var[\frac{\partial Cost}{\partial s^l}] = Var[\frac{\partial Cost}{\partial s^{l'}}]$, it ensures that backward propagation flows at a constant rate. Now, what remains is to calculate these variance values. An elaborate example has been given in the Appendix D.

**PEF saturation:** The derivative of the PEF is represented by: $= \frac{\alpha}{x^2}EF^2$. While the derivatives of the sigmoid and tanh are dependent on x, PEF is dependent on both $\alpha$ and x. Even if $\frac{EF^2(x)}{x^2}$

saturates, the learned parameter $\alpha$ will help the PEF escape saturation. The derivatives of the sigmoid, tanh saturate when $x > 5$ or $x < -5$. However, it is not true with PEF as evident from fig 2a. As empirical evidence, the layer values for every epoch of the model are captured using various activation functions like PEF, sigmoid and tanh. It is observed that, after about 10 epochs, the values of the layers becomes more or less constant for sigmoid and tanh (fig 2c and fig 2d), indicating their values have already saturated whereas for PEF, variation can be seen till it reaches 50 epochs (fig 2b). This shows that in comparison to sigmoid and tanh as activation functions, PEF escapes saturation due to its learned parameter $\alpha$. *The parameter $\alpha$ in PEF* changes its value as the model trains over the training dataset while using PEF as the activation function. Since it is a self training parameter, it returns different values for different datasets at the end of training. These values have been documented in table 2 and plotted in fig 2e. Table 2 demonstrates the variations in $\alpha$ values across multiple datasets as these values get updated.

The outcome of such an approach saves the overhead of tuning the model and also opens up avenues for discovering essential features of not-so-popular AFs. The inherent idea is to consider a 'fixed-form' activation and parameterize it. The parameter is 'learned' via the backpropagation step of the LSTM network such that the shape of the activation, determined by the parameter, is learned from data. Thus, if the dataset changes, so does the final form of the activation. In our case, the fixed form activation is the Elliot Activation function.

## 4 EXPERIMENT

In this section, we have evaluated the performance of the quantile-LSTM techniques on multiple datasets. We have identified multiple baseline methods, such as Isolation Forest (iForest), Elliptic envelope, Autoencoder and several deep learning based approaches for comparison purposes (See section 5 for more details on baseline methods).

### 4.1 DATASETS

The dataset properties have been shown in table 6. A total of 29 datasets, including real industrial datasets and synthetic datasets, have been considered in the experiments. The industrial datasets include Yahoo webscope (Yahoo!, 2019), AWS cloudwatch (Lavin & Ahmad, 2015), GE, etc. There are a couple of datasets with either one or few anomalies, such as $AWS_1$, $AWS_2$. We have injected anomalies in AWS, Yahoo, and GE datasets to produce synthetic data for fair comparison purposes. The datasets are univariate, unimodal or binodal and follow mostly Weibull, Gamma and Log normal distribution. The highest anomaly percentage is 1.47 (GE Dataset$_2$), whereas AWS Dataset$_2$ has reported the lowest percentage of anomaly i.e. 0.08 (For more details see Table 6 of Appendix, section E).

### 4.2 RESULTS

Table 3 demonstrates the performance comparison of various LSTM techniques. Precision and Recall, two performance metrics, are shown in the table. The Median-LSTM has achieved Recall 1 in most datasets (10 out of 15 datasets). In comparison to existing benchmarks, LSTM methods are SOTA on most of the datasets in terms of Recall. For comparison purposes, we have first compared the Recall. If the Recall is the same for two different methods, then we have compared the Precision. The method which has a higher Recall and Precision will be considered as a better performer. In AWS datasets, most of the techniques have achieved the highest Recall apart from DAGMM and DevNet. DevNet needs minimum two anomalies hence it is not applicable for AWS1 and AWS2. However, as per Precision, iqr-LSTM has performed better than other methods. In the case of GE1, DevNet has produced a better result, whereas quantile based LSTM techniques has outperformed others on GE$_2$. Median-LSTM has demonstrated better result in Ambient temperature. In the case of Yahoo datasets, Median-LSTM has achieved the highest Recall on four datasets; however, quantile-LSTM and iqr-LSTM have produced better results on several datasets. For example, Median-LSTM and iqr-LSTM both achieved Recall 1 on Yahoo$_1$. However, if we compare the Precision, iqr-LSTM has shown better results. It is evident from the table 3 that all these LSTM versions are performing very well on these industrial datasets.

| Dataset | Anomaly | iqr-LSTM | | Median-LSTM | | quantile-LSTM | | Autoencoder | | GAN | | DAGMM | | DevNet | | iForest | | Envelope | |
|---|---|---|---|---|---|---|---|---|---|---|---|---|---|---|---|---|---|---|---|
| | | Precision | Recall | Precision | Recall | Precision | Recall | Precision | Recall | Precision | Recall | Precision | Recall | Precision | Recall | Precision | Recall | Precision | Recall |
| AWS1 | 1 | 0.5 | 1 | 0.052 | 1 | 0.041 | 1 | 0.045 | 1 | 0.047 | 1 | 0.125 | 1 | NA | NA | 0.0087 | 1 | 0.009 | 1 |
| AWS2 | 2 | 0.13 | 1 | 0.22 | 1 | 0.0042 | 1 | 0.1 | 0.5 | 0.18 | 1 | 0.11 | 1 | NA | NA | 0.0062 | 1 | 0.04 | 1 |
| AWS3 | 1 | 1 | 1 | 0.37 | 1 | 0.0181 | 1 | 0.0344 | 1 | 0.055 | 1 | 0 | 0 | NA | NA | 0.005 | 1 | 0.006 | 1 |
| Ambient temperature | 1 | 0.03 | 1 | 0.0769 | 1 | 0.02 | 1 | 0.055 | 1 | 0 | 0 | 0 | 0 | NA | NA | 0.01 | 1 | 0.02 | 1 |
| GE1 | 3 | 0.019 | 1 | 0.048 | 1 | 0.0357 | 1 | 0.093 | 1 | 0.041 | 0.33 | 0 | 0 | 0.12 | 1 | 0.004 | 1 | 0.2 | 1 |
| GE2 | 8 | 1 | 1 | 0.66 | 1 | 1 | 1 | 1 | 1 | 0 | 0 | 0.8 | 1 | 0.8 | 1 | 0.16 | 1 | 0.034 | 1 |
| Yahoo1 | 2 | 0.076 | 1 | 0.0363 | 1 | 0.0465 | 1 | 1 | 0.5 | 0.066 | 1 | 0.07 | 0.5 | 0 | 0 | 0.005 | 1 | 0.009 | 1 |
| Yahoo2 | 8 | 0.75 | 0.375 | 0.8 | 1 | 1 | 0.375 | 1 | 0.25 | 0.19 | 0.625 | 0.10 | 0.25 | 0 | 0 | 0.04 | 0.875 | 0.055 | 1 |
| Yahoo3 | 8 | 0.615 | 1 | 0.114 | 0.675 | 0.088 | 1 | 0.023 | 0.25 | 0.11 | 0.875 | 0.15 | 0.62 | 0.39 | 0.5 | 0.04 | 0.875 | 0.032 | 0.875 |
| Yahoo5 | 9 | 0.048 | 0.33 | 0.1 | 0.33 | 0.022 | 0.66 | 0.05 | 0.33 | 0 | 0 | 0.23 | 0.33 | 0.67 | 1 | 0.029 | 0.66 | 0.029 | 0.66 |
| Yahoo6 | 4 | 0.12 | 1 | 0.222 | 1 | 0.0275 | 1 | 0.048 | 1 | 0 | 0 | 0.041 | 1 | 1 | 1 | 0.0073 | 1 | 0.0075 | 1 |
| Yahoo7 | 11 | 0.096 | 0.54 | 0.16 | 0.63 | 0.066 | 0.54 | 0.083 | 0.45 | 0.035 | 0.54 | 0.058 | 0.09 | 0.33 | 0.29 | 0.0082 | 0.33 | 0.017 | 0.54 |
| Yahoo8 | 10 | 0.053 | 0.7 | 0.142 | 0.8 | 0.028 | 0.3 | 0 | 0 | 0 | 0 | 0 | 0 | 0.063 | 0.11 | 0.01 | 0.6 | 0.010 | 0.6 |
| Yahoo9 | 8 | 1 | 0.75 | 0.333 | 1 | 0.0208 | 0.75 | 1 | 0.37 | 0 | 0 | 0.5 | 0.375 | 0.07 | 0.8 | 0.04 | 1 | 0.047 | 1 |

Table 3: Performance comparison of various quantile LSTM techniques with other state of the art algorithms.

| Dataset | Anomaly | iqr-LSTM | | Median-LSTM | | quantile-LSTM | | iForest | | Envelope | | Autoencoder | | GAN | | DAGMM | | DevNet | |
|---|---|---|---|---|---|---|---|---|---|---|---|---|---|---|---|---|---|---|---|
| | | Precision | Recall | Precision | Recall | Precision | Recall | Precision | Recall | Precision | Recall | Precision | Recall | Precision | Recall | Precision | Recall | Precision | Recall |
| AWS_syn1 | 11 | 0.769 | 0.909 | 0.687 | 1 | 1 | 0.909 | 0.034 | 1 | 0.10 | 1 | 1 | 0.63 | 0.84 | 1 | 0.71 | 0.90 | 0.09 | 0.73 |
| AWS_syn2 | 22 | 0.7 | 1 | 0.733 | 1 | 0.6875 | 1 | 0.065 | 1 | 0.33 | 1 | 0.5 | 0.63 | 0.70 | 1 | 0.56 | 1 | 0.44 | 0.27 |
| AWS_syn3 | 11 | 1 | 0.9 | 0.47 | 1 | 1 | 1 | 0.025 | 1 | 0.072 | 1 | 0.64 | 0.5 | 0.68 | 1 | 0 | 0 | 0.2 | 0.45 |
| GE_syn1 | 13 | 0.0093 | 1 | 0.203 | 1 | 0.071 | 0.769 | 0.0208 | 1 | 0.135 | 1 | 0.23 | 0.11 | 0.25 | 0.61 | 0 | 0 | 0.33 | 1 |
| GE_syn2 | 18 | 0.0446 | 1 | 1 | 1 | 1 | 1 | 1 | 1 | 0.3 | 1 | 1 | 0.38 | 0.9 | 1 | 0.9 | 1 | 0.9 | 1 |
| Yahoo_syn1 | 12 | 1 | 1 | 0.217 | 0.833 | 0.375 | 1 | 0.027 | 1 | 0.056 | 1 | 1 | 0.83 | 0.31 | 1 | 0.29 | 0.41 | 0 | 0 |
| Yahoo_syn2 | 18 | 0.181 | 0.55 | 0.653 | 0.944 | 1 | 0.611 | 0.233 | 1 | 0.124 | 1 | 1 | 0.42 | 1 | 0.61 | 0.55 | 0.61 | 0 | 0 |
| Yahoo_syn3 | 18 | 0.89 | 0.94 | 0.3333 | 0.555 | 0.6 | 1 | 0.0410 | 1 | 0.0762 | 0.944 | 1 | 0.88 | 0.81 | 0.71 | 0.3 | 0.66 | 0.17 | 0.63 |
| Yahoo_syn5 | 19 | 0.081 | 0.52 | 0.521 | 0.631 | 0.0625 | 0.578 | 0.03125 | 0.842 | 0.0784 | 0.842 | 0.15 | 0.47 | 0.42 | 0.53 | 0.52 | 0.52 | 0.73 | 0.92 |
| Yahoo_syn6 | 14 | 0.065 | 0.85 | 0.65 | 0.928 | 0.764 | 0.928 | 0.01825 | 1 | 0.00761 | 0.285 | 0.05 | 0.28 | 0.8 | 0.29 | 0.041 | 0.28 | 0 | 0 |
| Yahoo_syn7 | 21 | 0.61 | 0.59 | 0.375 | 0.714 | 0.411 | 0.66 | 0.032 | 0.952 | 0.052 | 0.85 | 0.18 | 0.42 | 0.14 | 0.38 | 0.058 | 0.047 | 0.11 | 0.64 |
| Yahoo_syn8 | 20 | 0.32 | 0.65 | 0.482 | 0.823 | 0.197 | 0.7 | 0.0192 | 0.75 | 0.023 | 0.7 | 0.009 | 0.05 | 0.25 | 0.1 | 0 | 0 | 0.23 | 0.64 |
| Yahoo_syn9 | 18 | 1 | 0.77 | 1 | 1 | 1 | 0.94 | 0.053 | 1 | 0.048 | 1 | 0.875 | 0.388 | 0.72 | 1 | 0.57 | 0.22 | 0.03 | 0.29 |

Table 4: Performance comparison of various quantile LSTM techniques on synthetic datasets with other state of the art algorithms.

Table 4 shows the comparison with other baseline algorithms on multiple synthetic datasets. As in the previous table, Recall and Precision have been shown as performance metrics. As per these metrics, quantile-based approaches have outperformed iForest and other deep learning based algorithms on 7 out of 13 datasets. If we consider the Precision alone, the quantile LSTM based techniques have demonstrated better performance on 10 synthetic datasets. Our methods outperformed tree based anomaly identifiers Multi-Generations Tree (MGTree) and Uni-variate Multi-Generations Tree (UVMGTree) (Sarkar et al., 2022) as well, in terms of Recall (See Table 8 of Appendix G). There are multiple reasons for the better performance demonstrated by the quantile based LSTM approaches. First is the efficacy of the LSTM, which is well documented. Median-LSTM has detected the anomalies for each time period utilizing mean and standard deviation. It also has helped to capture the trend and seasonality. quantile-LSTM do not have any predefined threshold, which has improved their performance. Additionally, the flexibility of the parameter $\alpha$ in determining the shape of the activation helped in isolating the anomalies. This is evident from Fig 2e which represents the variation in $\alpha$ values of the PEF function across the datasets. $\alpha$ has been initialized to $1.5$ for all the datasets. We have also experimented the algorithms on another well known benchmark dataset VLDB Paparrizos et al. (2022) (See Appendix M) and observed superior performance of median-LSTM over other benchmark methods including LSTM autoencoders where reconstruction loss in the autoencoder is used in conjunction with the recurrent structure to flag anomalies Srivastava et al. (2015).

## 5 RELATED WORK

Literature on quantile based anomaly detection (Tambuwal & Neagu, 2021; Solovjovs et al., 2021) including a quantile based Autoencoder approach (Ryu et al., 2022) suggested diversifying the

source of anomaly score by considering uncertainty term along with reconstruction error in anomaly score computation. However, there is no discussion on setting the appropriate threshold on the anomaly scores. Well known supervised machine learning approaches such as Linear Support Vector Machines (SVM), Random Forest (RF) and Random Survival Forest (RSF) (Voronov et al., 2018; Verma et al., 2017) have been explored for fault diagnosis and the life-time prediction of industrial systems. Popular unsupervised approaches such as Anomaly Detection Forest (Sternby et al., 2020), and k-means based Isolation Forest (Karczmarek et al., 2020) try to isolate the anomalies from the normal dataset. These methods like Anomaly Detection Forest do not require labeled data requires a training phase with a subsample of the dataset under consideration. A wrong selection of the training subsample can cause too many false alarms. Recently, Deep Learning (DL) models based on auto-encoders, long-short term memory (Erfani et al., 2016; Zhou & Paffenroth, 2017) are increasingly gaining attention for anomaly detection. Yin et al. (2020) have proposed an integrated model of Convolutional Neural Network (CNN) and LSTM based auto-encoder for Yahoo Webscope time-series anomaly detection. For reasons unknown, Yin et al. (2020) have taken only one Yahoo Webscope data to demonstrate their approach's efficacy. TA stacked LSTM (Malhotra et al., 2015) is used for time series anomaly prediction, and the network is trained on a normal dataset. Hierarchical Temporal Memory (HTM) method has been applied recently on sequential streamed data and compared with other time series forecasting models (Osegi, 2021). The authors in Saurav et al. (2018) have performed online timeseries anomaly detection using deep RNN. The incremental retraining of the neural network allows to adopt concept drift across multiple datasets. There are various works (Morales-Forero & Bassetto, 2019; Sperl et al., 2020), which attempt to address the data imbalance issue of the anomaly datasets since anomalies are very rare and occur occasionally. Hence they propose semi-supervised approaches but cannot avoid the expensive dataset labeling. Some approaches (Zong et al., 2018) apply predefined thresholds, such as fixed percentile value to detect the anomalies. However, a fixed threshold value may not be equally effective on different domain datasets. Deep Autoencoding Gaussian Mixture Model (DAGMM) is an unsupervised DL based anomaly detection algorithm (Zong et al., 2018), where it utilizes a deep autoencoder to generate a low-dimensional representation and reconstruction error for each input datapoint and is further fed into a Gaussian Mixture Model (GMM). Deviation Network(DevNet) (Pang et al., 2019) is a novel method that harnesses anomaly scoring networks, Z-score based deviation loss and Gaussian prior together to increase efficiency for anomaly detection.

# 6 DISCUSSION AND CONCLUSION

In this paper, we have proposed multiple versions of the SoTA anomaly detection algorithms along with a forecasting based LSTM method. We have demonstrated that combining the quantile technique with LSTM can be successfully implemented to detect anomalies in industrial and non-industrial datasets (See Table 7 of Appendix F) without label availability for training. It has been shown that the performance of the baseline methods is sensitive to the predefined thresholds (See Appendix L), whereas quantile based thresholds are generic. We have also exploited the parameterized Elliot activation function and shown anomaly distribution against quantile values, which helps in deciding the quantile anomaly threshold. The design of a flexible form activation, i.e., PEF, also helps in accommodating variance in the unseen data as the shape of the activation is learned from data. PEF, as seen in Table 9 in Appendix H captures anomalies better than vanilla Elliot. The statistical significance of the performance is highlighted in Table 16. The quantile thresholds are generic and will not differ for different datasets. The proposed techniques have addressed the data imbalance issue and expensive training dataset labeling in anomaly detection. These methods are useful where data is abundant. Traditional deep learning based methods use classical conditional means and assume random normal distributions as the underlying structure of data. These assumptions make the methods vulnerable to capturing the uncertainty in prediction and make them incapable of modeling tail behaviours. Quantile in LSTM (for time series data) is a robust alternative that we leveraged in isolating anomalies successfully. The distribution-agnostic behavior of Quantiles tuned out to be a useful tool in modeling tail behavior and detecting anomalies. The proposed methods have a few drawbacks 1. quantile based LSTM techniques are applicable only on univariate datasets. 2. A few of the methods such as quantile-LSTM, iqr-LSTM have dependency on multiple thresholds.

## 7 ETHICS STATEMENT

The present paper discusses various anomaly detection techniques and mostly performed the experiments on publicly available datasets for reproducibility purposes. The paper empirically studies various behaviors of (mostly existing) algorithms on machine/systems data. Since our experiment do not involve human or animal subjects, it is unlikely to introduce any ethical or societal concerns.

## 8 REPRODUCIBILITY

We have performed the experiments mostly on the publicly available datasets (except for the GE dataset). We have given the links of the datasets in the main text. Code of the quantile based LSTM techniques are available in https://anonymous.4open.science/r/Quantile-LSTM-D840/README.md..

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

# A  PROBABILITY BOUND

| Dataset | $\mathcal{P}(\mathcal{E} > 0.95)$ | $\mathcal{P}(\mathcal{E} > 0.9)$ | $\mathcal{P}(\mathcal{E} > 0.75)$ | $\mathcal{P}(\mathcal{F} < 0.25)$ | $\mathcal{P}(\mathcal{F} < 0.10)$ |
|---|---|---|---|---|---|
| AWS Dataset$_1$ | 0 | 0.01 | 0.004 | 0 | 0 |
| AWS Dataset$_2$ | 0 | 0.1 | 0.1 | 0 | 0 |
| AWS Dataset$_3$ | 0 | 0.007 | 0.0032 | 0 | 0 |
| Yahoo Dataset$_1$ | 0 | 0.014 | 0.005 | 0 | 0 |
| Yahoo Dataset$_2$ | 0 | 0.105 | 0.062 | 0 | 0 |
| Yahoo Dataset$_3$ | 0 | 0.103 | 0.076 | 0 | 0 |
| Yahoo Dataset$_4$ | 0 | 0.014 | 0.0055 | 0 | 0 |
| Yahoo Dataset$_5$ | 0 | 0.043 | 0.016 | 0 | 0 |
| Yahoo Dataset$_6$ | 0 | 0.028 | 0.011 | 0 | 0 |
| Yahoo Dataset$_7$ | 0 | 0.047 | 0.018 | 0.0069 | 0.017 |
| Yahoo Dataset$_8$ | 0 | 0.011 | 0.004 | 0.016 | 0.041 |
| Yahoo Dataset$_9$ | 0 | 0.017 | 0.0069 | 0.011 | 0.029 |
| Sensor Dataset$_1$ | 0 | 0.0344 | 0.0135 | 0 | 0 |
| Sensor Dataset$_2$ | 0 | 0 | 0 | 0.013 | 0.033 |
| GE Dataset$_1$ | 0 | 0.003 | 0.002 | 0 | 0 |
| GE Dataset$_2$ | 0 | 0.05 | 0.042 | 0 | 0 |
| AWS Datasetsyn$_1$ | 0 | 0.08 | 0.035 | 0 | 0 |
| AWS Datasetsyn$_2$ | 0 | 0.08 | 0.035 | 0 | 0 |
| AWS Datasetsyn$_3$ | 0 | 0.1 | 0.1 | 0 | 0 |
| Yahoo Datasetsyn$_1$ | 0 | 0.074 | 0.034 | 0 | 0 |
| Yahoo Datasetsyn$_2$ | 0 | 0.21 | 0.15 | 0 | 0 |
| Yahoo Datasetsyn$_3$ | 0 | 0.13 | 0.11 | 0 | 0 |
| Yahoo Datasetsyn$_5$ | 0 | 0.08 | 0.036 | 0 | 0 |
| Yahoo Datasetsyn$_6$ | 0 | 0 | 0 | 0.025 | 0.062 |
| Yahoo Datasetsyn$_7$ | 0 | 0.047 | 0.018 | 0.03 | 0.076 |
| Yahoo Datasetsyn$_8$ | 0 | 0.034 | 0.015 | 0.026 | 0.051 |
| Yahoo Datasetsyn$_9$ | 0 | 0.017 | 0.0069 | 0.034 | 0.088 |
| Sensor Datasetsyn$_1$ | 0 | 0 | 0 | 0.108 | 0.39 |
| Sensor Datasetsyn$_2$ | 0 | 0 | 0 | 0.146 | 0.36 |
| GE DatasetSyn$_1$ | 0 | 0.017 | 0.0104 | 0 | 0 |
| GE DatasetSyn$_2$ | 0 | 0.11 | 0.096 | 0 | 0 |

Table 5: Various probability values on different quantile threshold parameters (See lemma 1 of the main paper ).

# B  LEMMA

**Lemma 1:**
For an univariate dataset $\mathcal{D}$, the probability of an anomaly $\mathcal{P}(\mathcal{A}) = \mathcal{P}(\mathcal{E} > \alpha_{high}) + \mathcal{P}(\mathcal{F} < \alpha_{low})$, where $\alpha_{high}, \alpha_{low}$ are the higher and lower level quantile thresholds respectively.

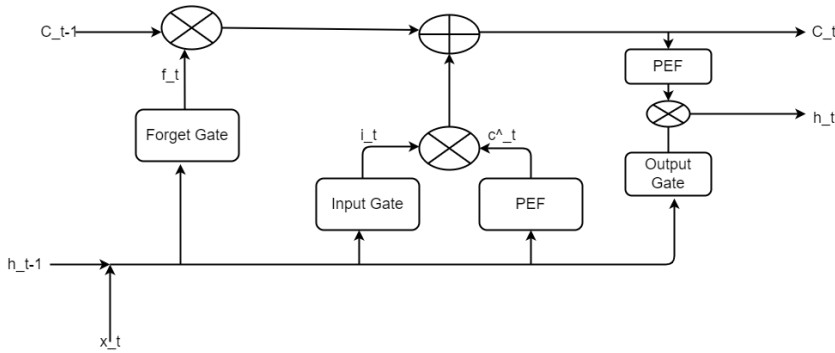

Figure 3: The structure of LSTM cell.

*Proof.* A datapoint is declared an anomaly if its quantile value is higher than $\alpha_{high}$ or lower than $\alpha_{low}$. Here $\alpha_{high}, \alpha_{low}$ are the higher and lower quantile threshold value. $\mathcal{P}(\mathcal{E} > \alpha_{high})$ is the probability of an anomaly whose quantile value is higher than $\alpha_{high}$. On the other side, $\mathcal{P}(\mathcal{F} < \alpha_{low})$ is the probability of quantile value of anomalous datapoint lower than $\alpha_{low}$. Hence the presence of an anomaly in a dataset is possible if one of the events is true. Therefore

$$\mathcal{P}(\mathcal{A}) = P(\mathcal{E} > \alpha_{high} \cup \mathcal{F} < \alpha_{low})$$
$$\mathcal{P}(\mathcal{A}) = \mathcal{P}(\mathcal{E} > \alpha_{high}) + \mathcal{P}(\mathcal{F} < \alpha_{low}) - P(\mathcal{E} > \alpha_{high} \cap \mathcal{F} < \alpha_{low})$$

Both the events $\mathcal{E}, \mathcal{F}$ are mutually exclusive. Hence the above Equation can be written as

$$\mathcal{P}(\mathcal{A}) = \mathcal{P}(\mathcal{E} > \alpha_{high}) + \mathcal{P}(\mathcal{F} < \alpha_{low}) \tag{2}$$

$\square$

## C  BACKPROPAGATION OF LSTM WITH PEF

For backward propagation, it is required to compute the derivatives for all major components of the LSTM. $J$ is the cost function and the relationship between $v_t$ and hidden state $h_t$ is $v_t = w_v * h_t + b_v$. The predicted value $y' = softmax(v_t)$. The derivative of the hidden state can be shown as follow

$$\frac{\partial J}{\partial h_t} = \frac{\partial J}{\partial v_t} \frac{\partial v_t}{\partial h_t}$$
$$\frac{\partial J}{\partial h_t} = \frac{\partial J}{\partial v_t} \frac{\partial(w_v * h_t + b_v)}{\partial h_t}$$
$$\frac{\partial J}{\partial h_t} = \frac{\partial J}{\partial v_t} w_v$$

The variable involved in the output gate is $o_t$.

$$\frac{\partial J}{\partial o_t} = \frac{\partial J}{\partial h_t} \frac{\partial h_t}{\partial o_t}$$
$$\frac{\partial J}{\partial o_t} = \frac{\partial J}{\partial h_t} \frac{\partial(o_t * PEF(C_t))}{\partial o_t}$$
$$\frac{\partial J}{\partial o_t} = \frac{\partial J}{\partial h_t} PEF(C_t)$$

$C_t$ is the cell state and the chain rule for cell state can be written as

$$\frac{\partial J}{\partial C_t} = \frac{\partial J}{\partial h_t} \frac{\partial h_t}{\partial C_t} \tag{3}$$

$\frac{\partial J}{\partial h_t}$ value already we have calculated as part of hidden state equation.

$$\frac{\partial h_t}{\partial C_t} = \frac{\partial(o_t * PEF(C_t))}{\partial C_t}$$
$$= \frac{\alpha o_t}{(|C_t| + 1)^2}$$

After setting the value of $\frac{\partial h_t}{\partial C_t}$ in equation 4

$$\frac{\partial J}{\partial C_t} = \frac{\partial J}{\partial h_t} \frac{\alpha o_t}{(|C_t| + 1)^2} \tag{4}$$

The chain rule for $\hat{c}_t$ is

$$\frac{\partial J}{\partial \hat{c}_t} = \frac{\partial J}{\partial C_t} \frac{\partial C_t}{\partial \hat{c}_t}$$

We need to derive only $\frac{\partial C_t}{\partial \hat{c}_t}$ since $\frac{\partial J}{\partial C_t}$ is available from equation 5.

$$\frac{\partial C_t}{\partial \hat{c}_t} = \frac{\partial(f_t * C_{t-1} + \hat{c}_t * i_t)}{\partial \hat{c}_t}$$
$$= i_t$$

After replacing the value of $\frac{\partial C_t}{\partial \hat{c}_t}$

$$\frac{\partial J}{\partial \hat{c}_t} = \frac{\partial J}{\partial C_t} * i_t$$

Similar way $\frac{\partial J}{\partial a_c} = \frac{\partial J}{\partial \hat{c}_t} * \frac{\alpha}{(|a_c|+1)^2}$ The following derivatives for input gate

$$\frac{\partial J}{\partial i_t} = \frac{\partial J}{\partial C_t} \hat{c}_t$$
$$\frac{\partial J}{\partial a_i} = \frac{\partial J}{\partial C_t} \hat{c}_t (i_t(1 - i_t))$$

For forget gate, below are the derivatives

$$\frac{\partial J}{\partial f_t} = \frac{\partial J}{\partial C_t} C_{t-1}$$
$$\frac{\partial J}{\partial a_f} = \frac{\partial J}{\partial C_t} C_{t-1}(f_t(1 - f_t))$$

$Z_t$ is the concatenation of the $h_{t-1}, x_t$. The derivatives of the weights are as follow

$$\frac{\partial J}{\partial w_f} = \frac{\partial J}{\partial a_f} Z_t$$
$$\frac{\partial J}{\partial w_i} = \frac{\partial J}{\partial a_i} Z_t$$
$$\frac{\partial J}{\partial w_v} = \frac{\partial J}{\partial v_t} h_t$$
$$\frac{\partial J}{\partial w_o} = \frac{\partial J}{\partial a_o} Z_t$$

## C.1 PARAMETERIZED ELLIOT FUNCTION

One of the major benefit of the parameterized Elliot function is that it further decreases the rate of saturation in comparison to the non-parameterize Elliot function. We have applied one parameter $\alpha$, which controls the shape of the Elliot function. There will be different derivatives if we apply parameterize Elliot function in LSTM.

$$PEF = \frac{\alpha x}{1 + |x|}$$

After the introduction of the PEF, the hidden state equation is as follow

$$h_t = O_t \alpha_c PEF(C_t)$$

As per the chain rule, the derivative for $\alpha_c$ will be

$$\frac{\partial J}{\partial \alpha_c} = \frac{\partial J}{\partial h_t} \frac{\partial O_t \alpha_c Elliot(C_t)}{\partial \alpha_c}$$

$$\frac{\partial J}{\partial \alpha_c} = \frac{\partial J}{\partial h_t} O_t * Elliot(C_t)$$

After each iteration, the $\alpha_c$ is updated as per equation 5.

$$\alpha_c^{(n+1)} = \alpha_c^n + \delta * \frac{\partial J}{\partial \alpha_c} \tag{5}$$

Similarly, we can derive $\hat{\alpha_c}$ and update the parameter.

## D  INTUITION WITH AN EXAMPLE:

Firstly, we attempt to find variance for two sigmoid activations in a network. The derivative of each activation output is approximately $0.25 (\sigma'(0) = 0.25)$, as the weights are uniformly initialized, and the input features are assumed to have the same variance. Hence,

$$f'(s_k^l) = 0.25$$

$$Var[z^2] = Var[x]((0.25)^2 n_1 Var[W^{1'}] * (0.25)^2 n_2 Var[W^{2'}])$$

We see that this diminishing factor of $0.25^N$ steeply drops the variance during the forward pass. Similarly, we observe that the gradient,

$$\frac{\partial Cost}{\partial s_k^l} = f'(s_k^l) W_{k,\cdot}^l \frac{\partial Cost}{\partial s^{l+1}}$$

has $f'(s_k^l)$ as one of the factors, and thus the diminishing factor is tied to the variance. Even when $N = 2$ the variance reduces by a factor of $4^4 = 256$.

Let's compute variance for neural network with two hidden layers using sigmoid and tanh activations. For tanh, if the initial values are uniformly distributed around 0, the derivative is $f'(s_k^l) = 1$. Therefore, the variance for the second layer output is, $Var[z^2] = Var[x] * ((0.25)^2 * n_1 * Var[W^{1'}] * n_2 * Var[W^{2'}])$. We see that the diminishing factor is just $4^2 = 16$, and this results in a much better variance when compared to the previous case. Therefore, using different AFs instead of the same implies a reduction in vanishing gradients and results in a much better flow of information because the variance value is preserved for longer.

# E    DATASET PROPERTIES

| Dataset Name | Anomaly% | Size | Missing Data | Modal | Distribution | #Variables | TW/ Period |
|---|---|---|---|---|---|---|---|
| Publicly available actual industrial data | | | | | | | |
| AWS Dataset$_1$ | 0.09 | 1049 | No | Unimodal | Weibull | Univariate | 84/168 |
| AWS Dataset$_2$ | 0.08 | 2486 | No | Unimodal | Weibull | Univariate | 38/152 |
| AWS Dataset$_3$ | 0.066 | 1499 | No | Multimodal | Weibull | Univariate | 38/152 |
| Yahoo Dataset$_1$ | 0.14 | 1421 | No | Unimodal | Weibull | Univariate | 20/60 |
| Yahoo Dataset$_2$ | 0.54 | 1462 | No | Unimodal | Gamma | Univariate | 30/90 |
| Yahoo Dataset$_3$ | 0.55 | 1440 | No | Unimodal | Weibull | Univariate | 10/120 |
| Yahoo Dataset$_4$ | 0.28 | 1422 | No | Bimodal | Weibull | Univariate | 105/210 |
| Yahoo Dataset$_5$ | 0.63 | 1421 | No | Bimodal | Log-normal | Univariate | 24/72 |
| Yahoo Dataset$_6$ | 0.28 | 1421 | No | Multimodal | Weibull | Univariate | 74/148 |
| Yahoo Dataset$_7$ | 0.53 | 1680 | No | Unimodal | Weibull | Univariate | 125/250 |
| Yahoo Dataset$_8$ | 0.59 | 1680 | No | Unimodal | Log-normal | Univariate | 116/232 |
| Yahoo Dataset$_9$ | 0.47 | 1680 | No | Unimodal | Weibull | Univariate | 30/90 |
| Machine Temperature Dataset | 0.19 | 501 | No | Multimodal | Weibull | Univariate | 38/114 |
| Private actual industrial data | | | | | | | |
| GE Dataset$_1$ | 0.18 | 1609 | No | Unimodal | Exponential | Univariate | 117/234 |
| GE Dataset$_2$ | 1.47 | 544 | No | Multimodal | Weibull | Univariate | 50/150 |
| Publicly available synthetic industrial data | | | | | | | |
| AWS DatasetSyn$_1$ | 1.03 | 1059 | No | Unimodal | Exponential | Univariate | 84/168 |
| AWS DatasetSyn$_2$ | 0.877 | 2506 | No | Unimodal | Weibull | Univariate | 38/152 |
| AWS DatasetSyn$_3$ | 0.72 | 1509 | No | Unimodal | Gamma | Univariate | 38/152 |
| Yahoo DatasetSyn$_1$ | 0.83 | 1431 | No | Unimodal | Weibull | Univariate | 20/60 |
| Yahoo DatasetSyn$_2$ | 1.22 | 1472 | No | Unimodal | Gamma | Univariate | 30/90 |
| Yahoo DatasetSyn$_3$ | 1.24 | 1450 | No | Unimodal | Weibull | Univariate | 10/120 |
| Yahoo DatasetSyn$_4$ | 0.977 | 1432 | No | Bimodal | Weibull | Univariate | 105/210 |
| Yahoo DatasetSyn$_5$ | 1.32 | 1431 | No | Bimodal | Weibull | Univariate | 24/72 |
| Yahoo DatasetSyn$_6$ | 0.97 | 1431 | No | Multimodal | Weibull | Univariate | 74/148 |
| Yahoo DatasetSyn$_7$ | 1.12 | 1690 | No | Unimodal | Weibull | Univariate | 125/250 |
| Yahoo DatasetSyn$_8$ | 1.18 | 1690 | No | Multimodal | Log-normal | Univariate | 116/232 |
| Yahoo DatasetSyn$_9$ | 1.065 | 1690 | No | Bimodal | Weibull | Univariate | 30/90 |
| Private synthetic industrial data | | | | | | | |
| GE DatasetSyn$_1$ | 0.80 | 1619 | No | Unimodal | Log-normal | Univariate | 117/232 |
| GE DatasetSyn$_2$ | 3.30 | 554 | No | Multimodal | Exponential | Univariate | 50/150 |

Table 6: Anomaly Dataset Properties.

# F    COMPARISON ON NON-INDUSTRIAL DATASETS

Since, we used industrial datasets in the initial comparison with multiple deep learning based anomaly detection techniques, thus to provide further survey, here we use non-industrial datasets

(See Table 7). Here, Deviation Networks gives NA because it does not work for single anomaly containing datasets.

| Dataset | Anomaly | quantile-LSTM | | Autoencoder | | GAN | | DevNet | |
|---|---|---|---|---|---|---|---|---|---|
| | | Precision | Recall | Precision | Recall | Precision | Recall | Precision | Recall |
| TravelTime$_{387}$ | 3 | 0.011 | 0.67 | 1 | 0.33 | 0.024 | 0.33 | 0.01 | 0.33 |
| TravelTime$_{451}$ | 1 | 0.006 | 1 | 0 | 0 | 0.016 | 1 | NA | NA |
| Occupancy$_{6005}$ | 1 | 0.03 | 1 | 0 | 0 | 0.007 | 1 | NA | NA |
| Occupancy$_{t4013}$ | 2 | 0.06 | 1 | 0.438 | 0.5 | 0.014 | 0.5 | 0.02 | 1 |
| Speed$_{6005}$ | 1 | 0.014 | 1 | 0.103 | 1 | 0.009 | 1 | NA | NA |
| Speed$_{7578}$ | 4 | 0.086 | 1 | 0.792 | 1 | 0.2 | 0.9 | 0.16 | 0.75 |
| Speed$_{t4013}$ | 2 | 0.053 | 1 | 0.75 | 0.5 | 0.043 | 1 | 0.1 | 1 |

Table 7: Performance comparison of various quantile LSTM techniques on non-industrial datasets.

## G  COMPARISON WITH TREE BASED ALGORITHMS

In this section, we have compared the quantile-LSTM with other tree based algorithms, such as UVMGTree, MGTree (Sarkar et al., 2022). In table 8, two performance metrics, Precision and Recall have been considered. It has been noticed from the same table that quantile-LSTM has produced better Recall, compared with other tree based anomaly identifier algorithms. However, both the tree based anomaly identifier algorithms have better Precision in comparison to quantile-LSTM. Since, Recall has been given higher priority, quantile-LSTM is competitive enough with other tree based anomaly detection algorithms.

| Dataset | UVMGTree(K-means) | | MGTree(K-means) | | quantile-LSTM | |
|---|---|---|---|---|---|---|
| | Precision | Recall | Precision | Recall | Precision | Recall |
| AWS Dataset$_1$ | 100% | 100% | 100% | 100% | 5.2% | 100% |
| AWS Dataset$_2$ | 18% | 100% | 100% | 50% | 2.2% | 100% |
| Yahoo Dataset$_1$ | 15% | 100% | 50% | 50% | 3.6% | 100% |
| Yahoo Dataset$_2$ | 66% | 22% | 100% | 12% | 61% | 100% |
| Yahoo Dataset$_3$ | 87% | 87% | 100% | 12% | 11% | 87% |
| Yahoo Dataset$_5$ | 33% | 33% | 100% | 33% | 10% | 33% |
| Yahoo Dataset$_6$ | 100% | 100% | 100% | 100% | 22% | 100% |
| Yahoo Dataset$_7$ | 77% | 63% | 50% | 27% | 16% | 63% |
| Yahoo Dataset$_8$ | 60% | 60% | 100% | 10% | 14% | 80% |
| Yahoo Dataset$_9$ | 100% | 100% | 100% | 62% | 33% | 100% |

Table 8: The performance comparison of UVMGTree with other standard anomaly identifier algorithms

## H    PERFORMANCE COMPARISON BETWEEN ELLIOT FUNCTION AND PEF AS ACTIVATION FUNCTION

In order to compare performance of Elliot function and parameterized Elliot function as activation functions, we experimented them by using them as activation functions in the LSTM layer of the models and comparing the results after they run on multiple datasets. The results are shown in Table 9.

| Dataset | Elliot Function | | Parameterized Elliot Function | |
|---|---|---|---|---|
| | Precision | Recall | Precision | Recall |
| AWS Dataset$_1$ | 0 | 0 | 0.041 | 1 |
| AWS Dataset$_2$ | 0.002 | 1 | 0.0042 | 1 |
| AWS Dataset$_3$ | 0.04 | 1 | 0.0181 | 1 |
| AWS DatasetSyn$_1$ | 0.02 | 0.73 | 1 | 0.909 |
| AWS DatasetSyn$_2$ | 0.39 | 0.77 | 0.6875 | 1 |
| AWS DatasetSyn$_3$ | 0.06 | 0.73 | 1 | 1 |
| Yahoo Dataset$_1$ | 0.006 | 0.25 | 0.0465 | 1 |
| Yahoo Dataset$_2$ | 0.02 | 1 | 1 | 0.375 |
| Yahoo Dataset$_3$ | 0.05 | 1 | 0.088 | 1 |
| Yahoo Dataset$_5$ | 0.001 | 0.33 | 0.022 | 0.66 |
| Yahoo Dataset$_6$ | 0.002 | 0.17 | 0.0275 | 1 |
| Yahoo Dataset$_7$ | 0.03 | 0.09 | 0.066 | 0.54 |
| Yahoo Dataset$_8$ | 0.017 | 0.4 | 0.028 | 0.3 |
| Yahoo Dataset$_9$ | 0.43 | 0.75 | 0.0208 | 0.75 |
| Yahoo DatasetSyn$_1$ | 0.14 | 0.86 | 0.375 | 1 |
| Yahoo DatasetSyn$_2$ | 0.04 | 0.72 | 1 | 0.611 |
| Yahoo DatasetSyn$_3$ | 0.1 | 0.78 | 0.6 | 1 |
| Yahoo DatasetSyn$_5$ | 0.004 | 0.31 | 0.0625 | 0.578 |
| Yahoo DatasetSyn$_6$ | 0.015 | 0.69 | 0.764 | 0.928 |
| Yahoo DatasetSyn$_7$ | 0.35 | 0.43 | 0.411 | 0.66 |
| Yahoo DatasetSyn$_8$ | 0.024 | 0.5 | 0.197 | 0.7 |
| Yahoo DatasetSyn$_9$ | 0.27 | 0.67 | 1 | 0.94 |

Table 9: Comparison of Precision and Recall score for LSTM with Elliot Function and parameterized Elliot Function as Activation Function.

According to the data gathered after running the models, we found that parameterized Elliot function has a better Precision and Recall for as except four of the datasets. Thus, we could conclude that using parameterized Elliot function as an activation function gave better performance for quantile-LSTM.

## I    BASELINE METHODS

We have identified multiple baseline methods, such as iForest, Elliptic envelope, Autoencoder and deep learning based approaches for comparison purposes. iForest is an isolation-based approach, whereas elliptic envelope is an unsupervised method.
**Isolation Forest**: iForest is an unsupervised anomaly detection algorithm Liu et al. (2008), which

creates a forest of binary trees based on random partition. It is an ensemble algorithm and the authors have shown that it is possible to isolate an anomaly with a minimum number of partitions. We have implemented the iForest using sklearn with the default configuration provided in the library.

**Elliptic Envelope** : It is another unsupervised approach which relies on a geometric configuration-ellipse to detect the anomalies. It creates an imaginary elliptic area on the datasets. The datapoints falling outside this elliptical area are considered as an anomaly. We have applied sklearn package of the python with default parameters to implement this algorithm.

**Autoencoder**: We have considered Autoencoder for comparison purpose. Like quantile-LSTM, Autoencoder is trained on the normal datapoints. It relies on the reconstruction error to identify any anomaly present in the dataset. We have computed the mean square error (MSE) to measure the reconstruction error. It detects the anomalies based on an upper and lower threshold on the MSE.

**Deep Autoencoding Gaussian Mixture Model (DAGMM)**: It is an unsupervised DL based anomaly detection algorithm Zong et al. (2018), where it utilizes a deep autoencoder to generate a low-dimensional representation and reconstruction error for each input datapoint and it is further fed into a Gaussian Mixture Model (GMM). Normal datapoints have been used for training purpose of the model. We have referred the implementation by Nakae Nakae (2018) and the configuration applied for KDDCUP datasets. However, we have implemented upper and lower threshold on the scores instead of percentile based threshold to detect the anomalies.

**Generative Adversarial Networks (GAN)** GANs are DL based generative models that generate samples similar to the training dataset by learning the true data distribution. It is an unsupervised approach and it is designed to find regularities or patterns in the input dataset in such way that it generates output very similar to the original dataset. The model consists of two sub-models, one is generator and another discriminator. The discriminator tries to identify the generated output either real or fake. We have considered the GAN implementation as a reference Sankar & Harper (2021). Like previous approach, we have applied upper and lower threshold instead of percentile based threshold to detect the anomalies.

**Deviation Network (DevNet)** This is a novel method that harnesses anomaly scoring networks, Z-score based deviation loss and Gaussian prior together to increase efficiency for anomaly detection. In this model first the anomaly scoring network is used to get a scalar anomaly score. Then the anomaly score is learned by a reference score generator to create a reference score. Lastly the reference score and its standard deviation are put into the deviation loss function to optimize the losses in the anomaly scores with reference to the mean of the reference scores.

**LSTM AutoEncoder** The LSTM autoencoder network architecture uses the first couple of neural network layers to create the compressed representation of the input data, the encoder. A repeat vector layer to distribute the compressed representational vector across the time steps of the decoder is then deployed. The final output layer of the decoder provides the reconstructed input data. Increase in reconstruction loss in the autoencoder is used to flag anomalies.

Autoencoder, GAN, DAGMM and DevNet all Deep Learning based baseline algorithms considered two thresholds ( upper threshold and lower thresholds) on reconstruction errors or predicted value. The thresholds are the same for all baseline algorithms. We have considered these two thresholds to align the baseline methods with the quantile based approaches. For example, in Autoencoder, upper and lower threshold are considered as follow:

$$Upper\_threshold = mean\_error + 2 * std\_error \qquad (6)$$

$$Lower\_threshold = mean\_error - 2 * std\_error \qquad (7)$$

,where $mean\_error$ and $std\_error$ are the mean and standard deviation of the reconstruction error. Equations 6 and 7 have been applied by all DL based baseline methods.

## J    F1 COMPARISON ON BENCHMARK DATASETS

Another metric which is very well-known for performance comparison is F1 scores. There are situations, where it is difficult to compare algorithms based on precision and recall. In those cases, F1 scores can be effective to evaluate the algorithms. Table 10 demonstrates the performance comparison of the quantile based LSTM techniques with other baseline algorithms in terms of F1 scores. It is evident from Table that quantile based LSTM approaches have outperformed others in most of the datasets.

| Dataset | iqr-LSTM | median-LSTM | quantile-LSTM | GAN | DAGMM | Autoencoder | LSTM-Autoencoders |
|---|---|---|---|---|---|---|---|
| AWS Dataset$_1$ | 0.66 | 0.1 | 0.8 | 0.09 | 0.22 | 0.86 | 0.057 |
| AWS Dataset$_2$ | 0.23 | 0.044 | 0.008 | 0.03 | 0.21 | 0.16 | 0.166 |
| AWS Dataset$_3$ | 1 | 0.071 | 0.035 | 0.1 | 0 | 0.066 | 0.035 |
| Yahoo Dataset$_1$ | 0.14 | 0.07 | 0.088 | 0.125 | 0.013 | 0.66 | 0.16 |
| Yahoo Dataset$_2$ | 0.5 | 0.88 | 0.54 | 0.29 | 0.14 | 0.4 | 0.4 |
| Yahoo Dataset$_3$ | 0.76 | 0.20 | 0.16 | 0.2 | 0.24 | 0.42 | 0.875 |
| Yahoo Dataset$_5$ | 0.084 | 0.15 | 0.042 | 0 | 0.27 | 0.086 | 0.125 |
| Yahoo Dataset$_6$ | 0.21 | 0.36 | 0.053 | 0 | 0.08 | 0.091 | 0.170 |
| Yahoo Dataset$_7$ | 0.16 | 0.25 | 0.11 | 0.066 | 0.071 | 0.14 | 0.101 |
| Yahoo Dataset$_8$ | 0.099 | 0.24 | 0.05 | 0 | 0 | 0 | 0.023 |
| Yahoo Dataset$_9$ | 0.85 | 0.5 | 0.04 | 0 | 0.42 | 0.54 | 0.533 |
| GE Dataset$_1$ | 0.038 | 0.09 | 0.06 | 0.074 | 0 | 0.17 | 0.20 |
| GE Dataset$_2$ | 1 | 0.8 | 1 | 0 | 0.088 | 1 | 1 |

Table 10: The F1 measure comparison of quantile based LSTM techniques with other standard anomaly identifier algorithms.

# K  CHARACTERISTICS OF THE LSTM TECHNIQUES

Here, we will discuss the strength and limitations of the approaches, proposed in this paper.

## K.1  QUANTILE-LSTM

The advantages of the quantile-LSTM are

1. It does not need the entire dataset in memory for anomaly identification.
2. Two thresholds (Upper, Lower) are used in quantile-LSTM and the thresholds are flexible.
3. It has exploited quantile, which is distribution agnostic.

However, it has several disadvantages.

1. This technique needs two LSTM models and two thresholds.
2. A special case of quantile-LSTM, iqr-LSTM requires three LSTM. Building multiple models can be expensive.

## K.2  MEDIAN-LSTM

The median-LSTM has a couple of advantages over other methods. They are listed as follow:

1. It has applied a single LSTM model, which computes the median unlike quantile-LSTM, which has applied two LSTM models.
2. median-LSTM does not identify the range of the normal data points; rather, it computes the distance of the observed value from the median.

The weakness of the approach are:

1. median-LSTM expects the entire time series dataset to be present from where it tries to identify the anomalies. In contrast, quantile-LSTM does not need the whole dataset to be present. Therefore, median-LSTM may require higher memory for huge number of datapoints.
2. In the case of quantile-LSTM, the upper and lower quantile values may differ for different datasets. However, it is always 0.5 quantile for median-LSTM irrespective of the datasets. Hence it is not that flexible like other approaches.

## L  VARYING THRESHOLDS

Autoencoder, GAN, DAGMM and DevNet all Deep Learning based baseline algorithms considered two thresholds ( upper threshold and lower thresholds) on reconstruction errors or predicted value. To understand the impact of different thresholds on the performance, we have considered three baseline algorithms GAN, Autoencoder and Devnet. The baseline methods have considered five different sets of threshold values for upper and lower thresholds. The sets are shown in column head of tables 11, 12 and 13, where the first threshold is the upper percentile/value and the second threshold is the lower percentile/value.

| GAN | 99.25 and 0.75 | | 99.75 and 0.25 | | 99.9 and 0.1 | | mean $\pm 1.5 * std$ | | mean $\pm 1 * std$ | |
|---|---|---|---|---|---|---|---|---|---|---|
| Dataset | Precision | Recall | Precision | Recall | Precision | Recall | Precision | Recall | Precision | Recall |
| Yahoo Dataset$_1$ | 0.09 | 1 | 0.25 | 1 | 0.5 | 1 | 0.0086 | 1 | 0.007 | 1 |
| Yahoo Dataset$_2$ | 0.348 | 1 | 0.333 | 0.375 | 0.4 | 0.25 | 0.037 | 0.875 | 0.013 | 1 |
| Yahoo Dataset$_3$ | 0.28 | 0.5 | 0.444 | 0.286 | 0.28 | 0.5 | 0.0543 | 0.5 | 0.0322 | 0.875 |
| Yahoo Dataset$_5$ | 0.007 | 1 | 0.007 | 1 | 0.007 | 1 | 0 | 0 | 0 | 0 |
| Yahoo Dataset$_6$ | 0.18 | 1 | 0.4 | 1 | 0.4 | 1 | 0 | 0 | 0 | 0 |
| Yahoo Dataset$_7$ | 0.154 | 0.364 | 0.3 | 0.273 | 0.5 | 0.182 | 0.041 | 0.63 | 0.04 | 0.63 |
| Yahoo Dataset$_8$ | 0.038 | 0.1 | 0.1 | 0.1 | 0.25 | 0.1 | 0 | 0 | 0 | 0 |
| Yahoo Dataset$_9$ | 0.192 | 0.625 | 0.5 | 0.625 | 0.5 | 0.25 | 0 | 0 | 0.0052 | 0.375 |

Table 11: Comparison of Precision and Recall score for GAN with varying thresholds for anomaly Upper Bound and Lower Bound(as mentioned in each column head).

| Autoencoders | 99.25 and 0.75 | | 99.75 and 0.25 | | 99.9 and 0.1 | | mean $\pm 1.5 * std$ | | mean $\pm 1 * std$ | |
|---|---|---|---|---|---|---|---|---|---|---|
| Dataset | Precision | Recall | Precision | Recall | Precision | Recall | Precision | Recall | Precision | Recall |
| Yahoo Dataset$_1$ | 0.5 | 0.07 | 0.5 | 0.036 | 0.5 | 0.019 | 0.0069 | 0.5 | 0.0099 | 1 |
| Yahoo Dataset$_2$ | 0.5 | 0.4 | 0.333 | 0.5 | 0.2 | 0.5 | 1 | 0.25 | 1 | 0.25 |
| Yahoo Dataset$_3$ | 0.44 | 0.5 | 0.4 | 0.5 | 0.25 | 0.333 | 0.7 | 0.875 | 0.122 | 0.875 |
| Yahoo Dataset$_5$ | 0.13 | 0.33 | 0.375 | 0.33 | 0.6 | 0.33 | 0.042 | 0.33 | 0.035 | 0.33 |
| Yahoo Dataset$_6$ | 0.18 | 1 | 0.5 | 0.33 | 0.5 | 0.5 | 0.023 | 1 | 0.022 | 1 |
| Yahoo Dataset$_7$ | 0.5 | 0.5 | 0.5 | 0.5 | 0.5 | 0.5 | 0.043 | 0.36 | 0.032 | 0.45 |
| Yahoo Dataset$_8$ | 0 | 0 | 0 | 0 | 0.25 | 0.1 | 0.008 | 0.1 | 0.004 | 0.1 |
| Yahoo Dataset$_9$ | 0.18 | 0.625 | 0.5 | 0.625 | 0.33 | 0.125 | 1 | 0.5 | 0.12 | 0.625 |

Table 12: Comparison of Precision and Recall score for Autoencoders with varying thresholds for anomaly Upper Bound and Lower Bound(as mentioned in each column head).

It is evident from the above tables that performance vary significantly based on the thresholds decided by the algorithm. Therefore it is very important to decide a correct threshold which can identify all the probable anomalies from the dataset. Most of the cases, it is not feasible to know the appropriate threshold before applying the baseline algorithms.

## M  VLDB

Using the VLDB Benchmark, we generated a timeseries dataset of 5500 datapoints containing 40 anomalies. We used various Deep Learining based algorithms on this generated non-industrial dataset. From table 14, we can clearly see that in terms of recall our proposed algorithm(Median-LSTM) works almost as well as GAN and far better than the other two. While in terms of precision Median-LSTM gives the best values, thus performing better than the other algorithms overall.

| Devnet | 99.25 and 0.75 | | 99.75 and 0.25 | | 99.9 and 0.1 | | mean $\pm 1.5 * std$ | | mean $\pm 1 * std$ | |
|---|---|---|---|---|---|---|---|---|---|---|
| Dataset | Precision | Recall | Precision | Recall | Precision | Recall | Precision | Recall | Precision | Recall |
| Yahoo Dataset$_1$ | 0.002 | 1 | 0.002 | 1 | 0.001 | 1 | 0 | 0 | 0.004 | 1 |
| Yahoo Dataset$_2$ | 0.005 | 1 | 0.005 | 1 | 0.005 | 1 | 0 | 0 | 0.004 | 0.4 |
| Yahoo Dataset$_3$ | 0.0078 | 1 | 0.0078 | 1 | 0.0078 | 1 | 0.0988 | 0.727 | 0.0354 | 0.909 |
| Yahoo Dataset$_5$ | 0.111 | 0.5 | 0.333 | 0.5 | 0.333 | 0.5 | 0.0279 | 1 | 0.0153 | 1 |
| Yahoo Dataset$_6$ | 0.167 | 1 | 0.5 | 1 | 0.5 | 0.667 | 0.333 | 1 | 0.0347 | 1 |
| Yahoo Dataset$_7$ | 0.054 | 0.2 | 0.125 | 0.2 | 0.25 | 0.2 | 0.033 | 0.571 | 0.0106 | 0.714 |
| Yahoo Dataset$_8$ | 0 | 0 | 0 | 0 | 0 | 0 | 0 | 0 | 0 | 0 |
| Yahoo Dataset$_9$ | 0 | 0 | 0 | 0 | 0 | 0 | 0 | 0 | 0 | 0 |

Table 13: Comparison of Precision and Recall score for Devnet with varying thresholds for anomaly Upper Bound and Lower Bound(as mentioned in each column head).

| VLDB Dataset | Precision | Recall |
|---|---|---|
| Median-LSTM | 0.513 | 0.976 |
| GAN | 0.0072 | 1 |
| Autoencoders | 0.013 | 0.025 |
| Devnet | 0.0357 | 0.158 |
| LSTM-Autoencoders | 0.064 | 0.143 |

Table 14: Comparison of Precision and Recall score for VLDB generated dataset for various deep learning anomaly detection techniques.

| VLDB Dataset | F1-score |
|---|---|
| Median-LSTM | 0.673 |
| GAN | 0.0143 |
| Autoencoders | 0.0171 |
| Devnet | 0.0582 |
| LSTM-Autoencoders | 0.0884 |

Table 15: Comparison of F1-score for VLDB generated dataset for various deep learning anomaly detection techniques.

# N  TABLES

We have experimented each of the algorithms on AWS datasets 5 times and mean($\mu$) and std deviation($\sigma$) of the results have been computed and shown in Table 16. The superior results are marked as bold. Iqr-LSTM (a variant of quantile-LSTM) has outperformed others in AWS1 and AWS3.

| Dataset | iqr-LSTM | | GAN | | Autoencoder | | DAGMM | |
|---|---|---|---|---|---|---|---|---|
| | Precision($\mu \pm \sigma$) | Recall($\mu \pm \sigma$) | Precision($\mu \pm \sigma$) | Recall($\mu \pm \sigma$) | Precision($\mu \pm \sigma$) | Recall($\mu \pm \sigma$) | Precision($\mu \pm \sigma$) | Recall($\mu \pm \sigma$) |
| AWS Dataset$_1$ | **875±0.216** | **1±0** | 0.061±0.008 | 1±0 | 0.026±0.0111 | 1±0 | 0.031±0.054 | 0.25±0.43 |
| AWS Dataset$_2$ | 0.126±0.067 | 1±0 | **0.16±0.0275** | **1±0** | 0.14±0.076 | 0.62±0.21 | 0.0425±0.038 | 1±0 |
| AWS Dataset$_3$ | **1±0** | **1±0** | 0.039±0.008 | 1±0 | 0.017±0.01 | 1±0 | 0±0 | 0±0 |

Table 16: Statistical analysis of the algorithms on AWS datasets.

| Dataset | $\alpha$ after training | $\alpha$ initial value |
|---|---|---|
| AWS Dataset$_1$ | 1.612 | 0.1 |
| AWS Dataset$_2$ | 0.895 | 0.1 |
| AWS Dataset$_3$ | 1.554 | 0.1 |
| AWS DatasetSyn$_1$ | 1.537 | 0.1 |
| AWS DatasetSyn$_2$ | 0.680 | 0.1 |
| AWS DatasetSyn$_3$ | 1.516 | 0.1 |
| Yahoo Dataset$_1$ | 1.432 | 0.1 |
| Yahoo Dataset$_2$ | 1.470 | 0.1 |
| Yahoo Dataset$_3$ | 1.658 | 0.1 |
| Yahoo Dataset$_5$ | 1.686 | 0.1 |
| Yahoo Dataset$_6$ | 1.698 | 0.1 |
| Yahoo Dataset$_7$ | 1.725 | 0.1 |
| Yahoo Dataset$_8$ | 1.850 | 0.1 |
| Yahoo Dataset$_9$ | 1.640 | 0.1 |

Table 17: Different $\alpha$ values for each Dataset after the training.

