# OpenReview forum: "QUANTILE-LSTM: A ROBUST LSTM FOR ANOMALY DETECTION"
_ICLR.cc/2023/Conference — Submitted to ICLR 2023_

### Official Review · Reviewer_Qabh · 2022-10-23

**Confidence:** 3
**Correctness:** 4
**Technical Novelty And Significance:** 2
**Empirical Novelty And Significance:** 2
**Recommendation:** 5

**Clarity, Quality, Novelty And Reproducibility:**

I don’t find it particularly clear where the authors think that their work adds value to existing work. Is the main insight that methods trained to learn quantiles are particularly good at detecting anomalies, or that the new activation function is very useful for LSTMs that predict quantiles or simply to combine a loss function with a learner? I find it difficult to evaluate the novelty value of this work without a clearer description of where the value of the present work is.

**Strength And Weaknesses:**

The main ideas of the paper are intuitive and clear, but the authors could do a better job tying the paper together. There is a) no clear explanation of how the two main ideas of the paper are connected and b) a proper discussion of the strengths and weaknesses of the various proposed alternatives.

**Summary Of The Paper:**

The paper combines ideas in quantile regression and anomaly detection with LSTM networks. The authors further propose a new activation function as an alternative to more established activation functions for LSTMs and run a range of experiments comparing their newly proposed method to other approaches for anomaly detection.

**Summary Of The Review:**

Most of the individual contributions of the paper seem rather marginal and disconnected, wherefore they don’t add up to an altogether interesting contribution. I am willing to change my mind if the authors are able to make a concrete proposal on what method they think is best for specific anomaly detection problems.

---

### Official Review · Reviewer_ZaFe · 2022-10-23

**Confidence:** 4
**Correctness:** 2
**Technical Novelty And Significance:** 2
**Empirical Novelty And Significance:** 2
**Recommendation:** 3

**Clarity, Quality, Novelty And Reproducibility:**

Clarity is good. Novelty is also good however choices in this work might make this line of research completely obsolete (choice of using LSTMs, no proper evaluation using necessary datasets/methods/measures)

Code is provided to assist in reproducibility of the claims

**Strength And Weaknesses:**

Strengths

1. Anomaly detection over time-series data is a timely, important, and well-studied problem
2. Well-written and easy-to-follow - well motivated
3. Experimental results support the claims in the paper

Weaknesses

1. The focus on LSTMs is not justified
2. Missing relevant baselines
3. Missing relevant datasets
4. Missing relevant evaluation measures to assess accuracy

Comments:

The paper focuses on LSTM without sufficient justification. Why? Is there proof that LSTMs is the best for this task? A strong argument is needed in that direction and likely experimental results to support this claim

The parametrization of the activation function is interesting. It would be better though if evaluated across different architectures/settings to understand the true impact and if indeed this is the reason for the improved performance

The work unfortunately misses many recent advances in the area, which as a result make the comparisons/evaluations somewhat obsolete.

For example, TSB-UAD is a new benchmark in that space. VUS is a new family of measures for evaluating time-series anomaly detection methods. The work should perform an evaluation using the benchmarking datasets, the baselines to demonstrate that indeed LSTMs is the way to go, and use the new evaluation measures which solve many flaws in the current evaluations.

"Tsb-uad: an end-to-end benchmark suite for univariate time-series anomaly detection." Proceedings of the VLDB Endowment 15.8 (2022): 1697-1711.

"Volume under the surface: a new accuracy evaluation measure for time-series anomaly detection." Proceedings of the VLDB Endowment 15.11 (2022): 2774-2787.

**Summary Of The Paper:**

The paper focuses on LSTM architectures for time-series anomaly detection. In particular, the paper proposes different solutions for taking into consideration conditional quantities, which help in the identification of anomalies. In addition, instead of using existing activation functions, the work proposes to use a learnable/parametrized function. A comparison against multiple baselines and datasets demonstrates the efficacy of the proposed solutions.

**Summary Of The Review:**

Check above for the reasoning. The paper can have a significant impact if these issues are improved but for now seems somewhat as a rushed/preliminary work

---

### Official Review · Reviewer_bCty · 2022-10-24

**Confidence:** 3
**Correctness:** 2
**Technical Novelty And Significance:** 2
**Empirical Novelty And Significance:** 2
**Recommendation:** 3

**Clarity, Quality, Novelty And Reproducibility:**

Novelty: The activation function PEF is novel, but it's performance has not been compared with a baseline activation function.

Clarity: Lacks clarity in presentation of results.

**Strength And Weaknesses:**

1. There should be a set of experiments that compare the anomaly detection performance of different activation functions since the paper proposes an activation function (PEF) as a novel contribution. This seems to be missing.


2. Section 4.2: "For comparison purposes, we have first compared the Recall." -- This is not the correct or standard practice. An algorithm that simply reports every data point as an anomaly will have recall 1 and beat all other algorithms. The paper should instead compare algorithms on the basis of F1 scores which is a combination of recall and precision.


3. Table 3, 4: The text in these tables is too small and hard to interpret. Should highlight best algorithms as per statistical significance analysis. Also, need to show std. errors.


4. Section 5 Related Work: We need more discussion on existing quantile-regression based anomaly detection. A simple search on Google Scholar for "anomaly detection quantile" comes up with existing published papers. e.g. [r1] which also works with multivariate data, unlike the proposed algorithm in this paper. Relevant works (more than one if applicable) should be treated as additional benchmarks to be compared with in experiments. The paper should clearly explain what is the novel contribution in the current work in light of the existing literature. Note that contribution in point 1 of the abstract "... estimate conditional quantiles ..." is not novel. And the evidence for the effectiveness of contribution point 2 of the abstract is not strongly convincing.


References:
[r1] Tambuwal, Ahmad Idris, and Daniel Neagu. "Deep Quantile Regression for Unsupervised Anomaly Detection in Time-Series." SN Computer Science 2.6 (2021): 1-16.


**Summary Of The Paper:**

The paper presents an algorithm to estimate conditional quantiles for time-series predictions, and proposes a new parameterized Elliot activation function in LSTM gates. The conditional quantiles in predictions are used to identify anomalies.

**Summary Of The Review:**

The paper lacks literature review in the specific domain (quantile based anomaly detection) and the experiments have missed out comparison among different activation functions.

---

### Official Review · Reviewer_9RXE · 2022-10-24

**Confidence:** 4
**Correctness:** 3
**Technical Novelty And Significance:** 3
**Empirical Novelty And Significance:** 3
**Recommendation:** 6

**Clarity, Quality, Novelty And Reproducibility:**

Clarity:
The paper is well written, organized, and technically detailed to a satisfactory extent. The notation is clear and consistent throughout the paper.

Quality:
The design and justifications for the proposed quantile-based approaches are technically sound. The observations made regarding the improvements in anomaly detection performance introduced by the proposed approaches are empirically well supported. Overall, the work seems to be mature in terms of quality.

Novelty:
From a methodological perspective, the contribution of this work can be considered rather incremental. In essence, the authors used quantiles to summarize time series on a higher resolution and applied classical LSTM networks on those “aggregated” time series. The authors also claim that they propose a so-called parameterized Elliott activation function (PEF) which, although effective, is a scaled version of the original Elliott activation function with the difference that the scaling multiplier is learned from data.

Reproducibility:
The experiments were conducted mostly on datasets which are publicly available. The authors have also made their code available through an anonymized repository. That being said, reproducing the results of this work should be attainable.

**Details Of Ethics Concerns:**

Not applicable.

**Strength And Weaknesses:**

Strengths:

* This work is simple yet effective. Making use of simple statistics such as quantiles, the authors devise three LSTM-based architectures that exhibit state-of-the-art anomaly detection performance when compared to both well-established anomaly detection methods as well as anomaly detection approaches based on deep learning.

* The paper is well motivated by the distribution-agnostic properties of quantiles and their suitability for capturing tail behavior.

* One of the main advantages of the proposed approaches is their advantage over threshold-based anomaly detection methods. Namely, many anomaly detection methods rely on global thresholds for deciding whether a data point is anomalous or not, which are specific to the domain and the model being used (e.g., autoencoder-based approaches require determining a specific threshold for their reconstruction errors); and thus determining reasonable thresholds may be rather difficult. In contrast, relying on quantiles, the proposed approaches make use of adaptive, domain-invariant and dataset-invariant thresholds that do not need to be set beforehand.

* The proposed quantile-based approaches make no assumption about the data distribution. This is supported empirically by showing that the distributional variance does not impact the predictive performance of the approaches.

* The proposed PEF activation function satures slower/later than other activation functions such as sigmoid and tanh. Its saturation ratio is less pronounced and the authors demonstrate that such an activation function leads to improved anomaly detection performance.

* The scaling parameter $\alpha$ in PEF is learned automatically from data which allows for flexibility in determining the shape of PEF that is better suited for isolating anomalies. This is supported by the observation that different $\alpha$ values have been learned and have shown to be beneficial for the different datasets used in the experiments.

* The authors have conducted relatively thorough experiments to assess the effectiveness of the proposed approaches in comparison with the baseline methods.

* The paper is clear and fairly well-written. The used notation is easy-to-follow and consistent throughout the entire paper.

-------------------------------------------------------------------------------------------------

Weaknesses:

* Due to the large number of parameters that can be introduced, LSTM networks are arguably not always well-suited for (very) long sequences. In that regard, I am wondering if the proposed approaches are limited to short time series. I would encourage the authors to elaborate on this point in their response. Also, including the length of the times series for each of the datasets in the paper (or in the Appendix) would be beneficial.

* It is not clear whether the autoencoder baseline leverages LSTMs as encoder and decoder networks or regular feed-forward neural networks. If the latter is the case, I am wondering why the authors did not consider comparing to such a relevant baselines as an LSTM autoencoder for anomaly detection. I would like to ask if the authors can clarify this in their response as well as in the paper.

* Towards the end of Section 2.1.3, the authors discuss their choice of $\omega$. Namely, the authors claim that, when $\omega=2$, 95.45% of the data points are within two standard deviations distance from the mean value. Nevertheless, I am wondering how this parameter can be determined without making any assumptions about the data distribution? For example, for a certain dataset, the claimed 95.45% of the data points might fall within two standard deviations of the mean, but this might not be the case for another dataset. Thus, I would encourage the authors to include more specific details on the choice of $\omega$ and how this choice depends on the underlying distribution of the dataset at hand.

* Lemma 1 seems to be self-evident. Moreover, its proof appears to be rather trivial. Therefore, I am wondering if this lemma is necessary to be a part of the main paper. Moreover, I believe that this lemma would hold for “outliers”, not necessarily for “anomalies”, as data points that are not within a certain range of quantiles are typically referred to as “outliers” which can be data points that are distant from the mean or location of a distribution but not necessarily represent abnormal behavior. On the other hand, the notion that anomalies can also be declared by means of lower and higher quantiles is essentially the hypothesis made in this paper. It would be appreciated if the authors can provide some clarity on this point in their response.

* To assess the anomaly detection performance of the proposed approaches and the baselines, recall and precision have been calculated. Although the proposed quantile-based approaches are threshold-invariant, recall and precision must have been calculated at a particular threshold for the threshold-dependent baselines. Were different thresholds determined for each of these baselines or the same threshold was applied across all of them? I would suggest that the authors include in the paper the threshold(s) that was/were used for each of the datasets. Lastly, I believe that this work would have benefited from an analysis of the effect that different thresholds have on the performance of the baselines. Such an analysis would have verified whether the performance improvements hold in case different thresholds are applied to the baselines.

-------------------------------------------------------------------------------------------------

Minor weaknesses:
There are also certain grammatical and typographical errors and remarks that require attention. Some of them are summarized as follows:
- Page 2: “Our contributions are three folds” should be replaced with “Our contributions are three-fold”.
- Usage of informal language: “Thus, in time series data, where LSTM architecture has shown to come in handy”. In this sentence, consider replacing “come in handy” with “useful”, “practical” or “beneficial”.
- The second sentence on page 4 starts with “quantile-LSTM” which is not capitalized. Same applies to the first sentence in Section 2.1.2.
- A full stop is missing at the end of Section 2.1.2.
- Page 4, Section 2.1.3: “After computing the differences on the entire dataset, for every window $D_p$, mean ($\mu_p$) and standard deviation ($\sigma_p$) for the individual time period $D_p$.” —> In this sentence, the phrase “are calculated” should be inserted between “($\sigma_p$)” and “for”.
- Table 2 is included as a figure whereas it should be included as a table using the {table} environment.
- In the caption of Figure 3, “learn” should be replaced with “learnt” or “learned”.


**Summary Of The Paper:**

This paper proposes three different approaches to estimating quantiles from time series data using long short-term memory networks (LSTMs). In addition, a parameterized Elliott function (PEF) is proposed and used as an activation function in each of the quantile-based LSTM variants. When compared to several industrial time series datasets, quantile-based LSTMs outperform several conventional anomaly detection methods as well as deep learning models for anomaly detection.

**Summary Of The Review:**

Overall, this paper proposes a simple yet effective approach to anomaly detection that seems to outperform several state-of-the-art anomaly detection baselines. The work lacks methodological novelty and is rather incremental in that respect. Note that this does not diminish the importance of the performance improvements introduced by the proposed quantile-based approaches across several anomaly detection datasets. That being said, considering:

(1) the degree to which this work is application-oriented rather than a work that makes novel methodological contributions in representation learning, and

(2) the absence of a comparison of the proposed approaches to variants of the baselines run with different thresholds;

I am not convinced that this work is a good fit for a venue such as ICLR. Nevertheless, I am looking forward to the authors’ response and I would be willing to adjust my score in case I have misunderstood or misinterpreted certain aspects of the work.

---

### Decision · Program_Chairs · 2023-01-20

**Decision:**

Reject

**Justification For Why Not Higher Score:**

The idea is interesting but is explained without justification.

**Justification For Why Not Lower Score:**

N/A

**Metareview: Summary, Strengths And Weaknesses:**

This paper presents an anomaly detection method in times series, where the main idea is to devise three different LSTM models to predict quantiles in a future window. Quantile-based LSTMs seem to be sound and interesting. The parameterized Elliot function (PEF) is another interesting contribution in this paper. However, there are some concerns that were not resolved. First of all, it is not clear why quantile-LSTM is preferred over many existing methods. It would be nice to place a good motivation in introduction, so that readers are convinced. Some reviewers criticized many relevant works are missing. So this should be improved for future submissions. What is most important is to justify why learning quantiles using LSTMs is a good approach to anomaly detection. Therefore, the paper is not recommended for acceptance in its current form. I hope authors found the review comments informative and can improve their paper by addressing these carefully in future submissions.